# No Change Detected in Culturable Fungal Assemblages on Cave Walls in Eastern Canada with the Introduction of *Pseudogymnoascus destructans*

**Karen J. Vanderwolf** [1,2,*,†] **, David Malloch** [1] **and Donald F. McAlpine** [1]

1   New Brunswick Museum, Saint John, New Brunswick, NB E2K 1E5, Canada;
    dmalloch@xplornet.com (D.M.); Donald.McAlpine@nbm-mnb.ca (D.F.M.)
2   Canadian Wildlife Federation, Kanata, ON K2M 2W1, Canada
*   Correspondence: kjvanderw@gmail.com
†   Current address: Trent University, Peterborough, ON K9L 0G2, Canada.

**Abstract:** Studies of fungi in caves have become increasingly important with the advent of white-nose syndrome (WNS), a disease caused by the invasive fungus *Pseudogymnoascus destructans* (*Pd*) that has killed an estimated 6.5 million North American bats. We swabbed cave walls in New Brunswick, Canada, in 2012 and 2015 to determine whether the culturable fungal assemblage on cave walls changed after the introduction of *Pd* and subsequent decrease in hibernating bat populations. We also compared fungal assemblages on cave walls to previous studies on the fungal assemblages of arthropods and hibernating bats in the same sites. The fungal diversity of bats and cave walls was more similar than on arthropods. The diversity and composition of fungal assemblages on cave walls was significantly different among media types and sites but did not differ over time. Therefore, no change in the culturable fungal assemblage present on cave walls was detected with the introduction of *Pd* and subsequent disappearance of the hibernating bat population over a 3-year period. This suggests that fungi documented in caves in the region prior to the outbreak of *Pd* do not require regular transmission of spores by bats to maintain fungal diversity at these sites.

**Keywords:** cave fungi; culture media; *Pseudogymnoascus*; bat fungi; white-nose syndrome; speleology; cave microbiology; fungal ecology; cave biology

## 1. Introduction

Microscopic fungi are an important part of cave ecosystems. Fungi are found on a variety of substrates in caves but are not evenly distributed within underground environments. The highest diversity of fungi in caves are associated with deposits of organic material, such as dung [1,2]. Fungal spores can be transported into caves by water, wind, and fauna such as arthropods and bats. In fact, the majority of fungi documented from caves appear to originate from the non-subterranean environment since the fungal taxa most commonly reported from caves are also commonly found in the environment above ground [1], and fungal diversity decreases with increasing distance from entrances [3].

Despite the apparent lack of organic material, a diverse fungal assemblage has previously been documented on cave walls [3–6]. In [7], it was found that fungal diversity on cave walls differed from that on the floor of caves. Unlike the habitat of rock-inhabiting microbes outside caves, rock walls deep in caves are exposed to high humidity, never exposed to UV light, experience minimal temperature variation, and are sheltered from weather events such as wind and rain, all of which may affect the microbial community [8,9]. Bacteria in caves are known to acquire energy by transforming aromatic compounds, fixing gases such as methane, trapping particulate material from the atmosphere, and

oxidizing metals within rocks [8–10]. Fungal energy acquisition in caves is less well studied, although similar modes of nutrition have been reported on bare rock above ground [11,12]. The composition and abundance of fungal communities on bare rock outside caves is influenced by rock type and topography, with fungi favoring colonization of hollows on the rock surface [13]. The availability of energy sources for microbes in the atmosphere of caves is determined by convective cave ventilation and the presence of fauna and flora, such as dung and plant roots hanging from the ceiling [9]. Decomposing dung emits a variety of gases, including hydrogen sulfide, carbon dioxide, ammonia, and methane, and [14] noted that caves with large guano deposits have high levels of gaseous ammonia. These gases may be nutrition sources for microbes on cave walls. Fungi on cave walls may also obtain energy from nutrients in percolating water [3], or by oxidizing metals in the environment.

The study of fungi in caves has become increasingly important with the advent of white-nose syndrome (WNS), a disease caused by the fungus *Pseudogymnoascus destructans* (*Pd*; [15]) that has killed an estimated 6.5 million North American bats [16]. The effect of the introduction of this invasive fungus to North American cave ecosystems is unknown, although it has been documented on multiple substrates in caves including sediments [17], cave walls [18,19], and arthropods [20] in addition to bats. *Pd* may influence cave ecosystems through interactions with native microflora or by the removal of hibernating bats due to high mortality from WNS. Bats may act as vectors of fungal spores into caves [21,22] or provide energy sources for fungi in the form of guano or dead bats. In tropical caves with large guano deposits bats are thought to fuel cave food webs and influence the fungal assemblage [7,23], although this is less likely in temperate caves with little guano. While hibernating in underground environments, bats are in constant contact with cave walls or the ceiling, and therefore may influence, or be influenced by, the fungal assemblage on these areas.

WNS was first detected in New Brunswick, Canada in March 2011 and subsequently spread to all known hibernacula by April 2013 ([24,25]; unpublished data). The goals of this study were to 1) determine if the culturable fungal assemblage on cave walls differs from assemblages previously documented in the same sites on arthropods [20] and hibernating bats [21,22,26] and 2) establish whether the culturable fungal assemblage on cave walls changed after the introduction of *Pd* to the region and subsequent decrease in the hibernating bat population.

## 2. Materials and Methods

We collected swabs from hibernacula where bats (*Myotis lucifugus* and *M. septentrionalis*, hereafter *Myotis* spp. and *Perimyotis subflavus*) had been observed in southern New Brunswick, Canada. Data on physical characteristics of study sites, including location, hibernacula length, and seasonal temperatures are in [25]. We swabbed cave walls in three hibernacula, Dorchester Mine, Berryton Cave, and Glebe Mine, both in April 2012 and April 2015. We sampled an additional site, White Cave, in April 2015 because hibernating bats were still roosting there, unlike the other sites. We sampled fungal colonies observed growing on the walls of Kitts Cave in 2013 (Figure 1). Berryton Cave and Kitts Cave are Mississippian limestone, White Cave is Mississippian gypsum and anhydrite, Glebe Mine contains manganese, and Dorchester Mine contains copper deposits and minerals such as chalcocite, bornite, chalcopyrite, and azurite [27]. We surveyed sites for bats twice a year (late fall and late winter) from 2010 to 2015. We first saw bats with visible WNS infection in Berryton Cave March 2011, Dorchester Mine December 2011, White Cave December 2011, Kitts Cave February 2012, and Glebe Mine March 2012. We previously swabbed bats for fungi in all sites except Dorchester Mine in winter 2010 and no *Pd* was detected [21]. The date of *Pd* arrival in each site is uncertain since bats with WNS may not have visible *Pd* growth [28]. We followed the protocol of [29] for minimizing the spread of WNS during all visits to caves.

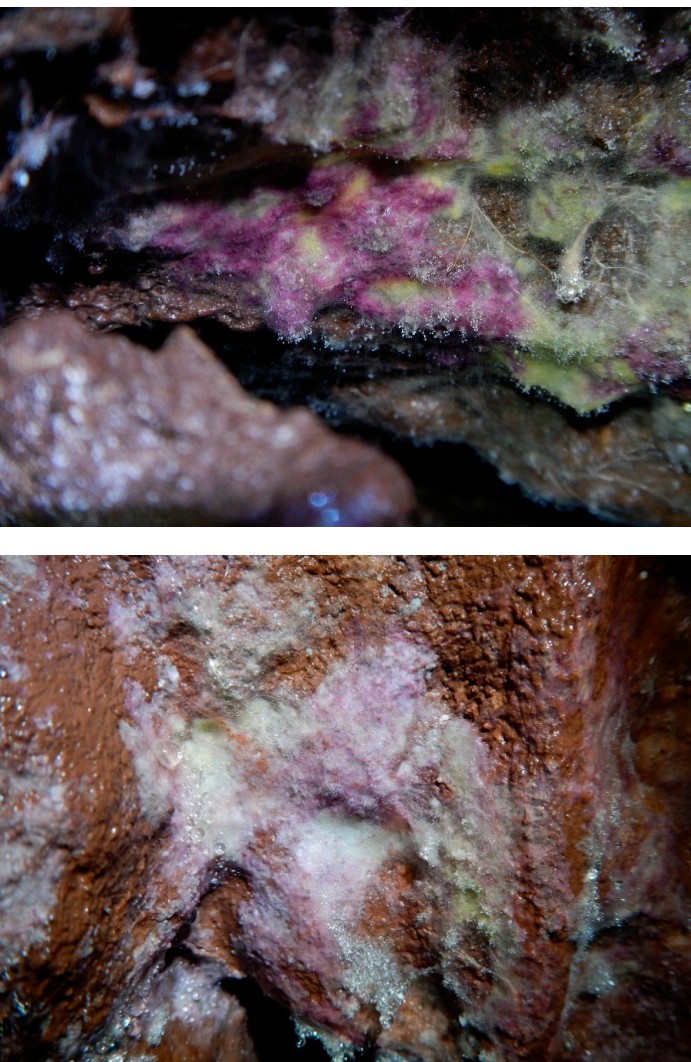

**Figure 1.** Fungi growing on cave walls covered in clay in Kitts Cave, New Brunswick, sampled April 2013.

We collected swabs ~2 m above the cave floor in the dark zone of hibernacula in areas where bats routinely roosted pre-WNS, but always >1 m from the nearest roosting bat. We collected samples from the same areas within hibernacula in 2012 and 2015. At each hibernaculum, we swabbed areas of approximately 20 × 20 cm on the wall during each visit, with one swab used for each 20 × 20 cm area. Methods were identical to those used to document fungi present on arthropods and bats at the same sites [20–22,26]. We took each sample with a sterile, dry, cotton-tipped applicator that was immediately streaked on the culture medium surface in a petri plate. We completed diluting streaks within hibernacula within 1 h of the initial streak, after which we sealed plates in situ with parafilm (Pechiney Plastic Packaging, Chicago, IL, USA). We used a new applicator for each wall swab. We shuffled plates in the field to ensure media types were not inoculated sequentially. In 2012, we used dextrose-peptone-yeast extract agar (DPYA; [30]) and Sabouraud-dextrose agar (SD), with 5 plates of each media type for Dorchester Mine and Glebe Mine, and 6 of each for Berryton Cave. In 2013, we swabbed growing fungal colonies in Kitts Cave and streaked swabs onto DPYA only. In 2015, we used DPYA, SD, MEA (malt-extract agar), and modified DPYA, with 5 plates of each media type for each of the four hibernacula. We added the antibiotics chlortetracycline (30 mg/L) and streptomycin (30 mg/L) to all media. Modified DPYA differed from DPYA in that we autoclaved the agar separately from the other ingredients. In [31], it was found that autoclaving ingredients separately increased the number of

bacterial species that could be cultured, and this could potentially apply to culturable fungal diversity as well. MEA consisted of 20 g Malt extract, 1 g peptone, 20 g dextrose, and 20 g agar/liter. DPYA consisted of: 5 g dextrose, 1 g peptone, 2 g yeast extract, 1 g $NH_4NO_3$, 1 g $K_2HPO_4$, 0.5 g $MgSO_4 \cdot 7H_2O$, 0.01 g $FeCl_3 \cdot 6H_2O$, 5 g oxgall, 1 g sodium propionate, and 20 g agar/L [30].

We incubated inverted plates in the dark at 7 °C, a temperature that approximates that found in our study sites [25], and monitored them over 4 months, in the manner of [21]. We maintained pure cultures of each distinct colony on DPYA without oxgall and sodium propionate. We identified isolates by comparing micro- and macromorphological characteristics of the microfungi to taxonomic literature and compendia [32,33], and by using a reference collection of cultures from *Myotis* spp. collected in 2010 that were previously identified using a mix of sequencing and morphological features [21]. Some isolates were sent to taxonomic specialists for confirmation of identification, usually through a combination of morphological and molecular genetic techniques. Permanent cultures are housed in the UAMH Center for Global Microfungal Biodiversity (UAMH 11722, 11729) and desiccant dried samples are vouchered in the New Brunswick Museum (NBM#F-05200–05245, 05262–05290, 05301–05302, 05312, 05314–05348, 05363, 05402–05423, 05428–05430, 05434, 05455–05520, 05529–05535, 05577–05625).

*Statistical Analysis*

Data on fungi cultured from bats and arthropods in the same caves that walls were swabbed were taken from [20–22,26]. We conducted the arthropod study winters 2012–2014 with Cave Orb Weavers (*Meta ovalis*), harvestmen (*Nelima elegans*), Herald Moths (*Scoliopteryx libatrix*), and fungus gnats (*Exechiopsis* sp. with a few *Anatella* sp.). We conducted the bat studies winters 2010, 2012, and 2013 with *M. lucifugus*, *M. septentrionalis*, and *Perimyotis subflavus*. In these previous studies we used four plates per bat and two plates per arthropod, except for *Exechiopsis* spp. which had 3–4 individuals per plate. In cases of multiple plates per individual we only used data from one DPYA plate per bat and per arthropod in the analysis. We performed all analyses in R [34]. The Akaike information criterion (AIC) indicated that, among potential response variables, the Simpson diversity index produced the best model compared to species richness and Shannon index. We calculated the Simpson diversity index for each sample, transformed it by squaring, and used it as the response variable in an ANOVA with substrate, site, and year as independent variables. Due to high stress values in non-metric dimensional scaling (NMDS) plots, we averaged data for each substrate type in each site in each year. We performed a non-parametric permutational multivariate analysis of variance (PERMANOVA) with 999 permutations using the function ADONIS in the vegan package [35] and NMDS plots using Bray–Curtis dissimilarity coefficients on the averaged data. We tested differences in multivariate dispersion between groups using the betadisper function (vegan package). We report the $R^2$ values (amount of community variance explained by the variable) when the variable enters the model last. We performed an indicator species analysis to identify specific fungal taxa associated with substrate type using the multipatt function in the indicspecies package [36]. We adjusted *p*-values for multiple tests with the FDR method (false discovery rate).

We tested the effect of media type, year, and site on the fungal diversity cultured from caves walls using an ANOVA after checking assumptions. We performed a PERMANOVA on Jaccard similarity coefficients to examine differences among fungal assemblages using site, media type, and year as explanatory factors. We constructed NMDS plots of Bray–Curtis dissimilarity coefficients to visualize how different sites and media types cluster. We performed an indicator species analysis to identify specific fungal taxa associated with different sites and media types.

## 3. Results

*3.1. Fungal Diversity on Cave Walls Compared to Arthropods and Bats*

The Simpson diversity index of fungi per sample varied significantly with substrate ($F_{7,25} = 9.017$, $p = 1.56 \times 10^{-5}$) and site ($F_{4,25} = 3.994$, $p = 0.012$) but not year sampled ($F_{1,25} = 0.425$, $p = 0.520$). Cave

walls generally had the highest fungal diversity within a site compared to arthropods, while overall, bats had the lowest diversity (Figure 2). The composition of fungal taxa on bats, arthropods, and cave walls significantly varied by site (F.model$_{4,25}$ = 1.878, R$^2$ = 0.141, $p$ = 0.001), and particularly substrate (F.model$_{7,25}$ = 2.241, R$^2$ = 0.295, $p$ = 0.001), but not by year sampled (F.model$_{1,25}$ = 0.818, R$^2$ = 0.015, $p$ = 0.695). All variables had significant dispersion (site: F$_{4,33}$ = 3.751, $p$ = 0.016; substrate: F$_{7,30}$ = 4.957, $p$ = 0.003; year: F$_{4,33}$ = 3.810, $p$ = 0.017). The fungal diversity of bats and cave walls were more similar compared to fungal diversity on arthropods (Figure 3A). Sites did not cluster, except for Berryton Cave and White Cave (Figure 3B). Glebe Mine and Dorchester Mine are distinct from the other sites (Figure 3B). We did not detect any significant indicator species by substrate, although multiple species of fungi were often associated with walls as opposed to bats or arthropods (Table 1), and some fungal taxa were only associated with arthropods.

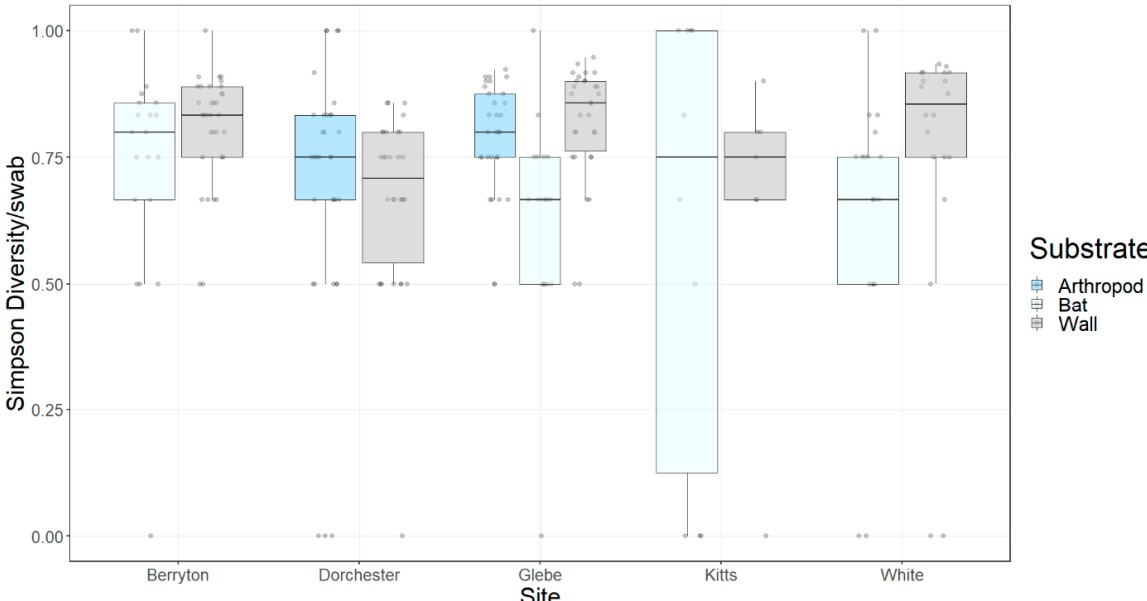

**Figure 2.** The transformed Simpson diversity index of fungi per swab (represented with dots) on different substrates in five caves in New Brunswick, Canada. The boxes represent the mean, the first and third quartiles, and the whiskers extend from the box to the value no further than ± 1.5 multiplied by theinter-quartile range from the boxes.

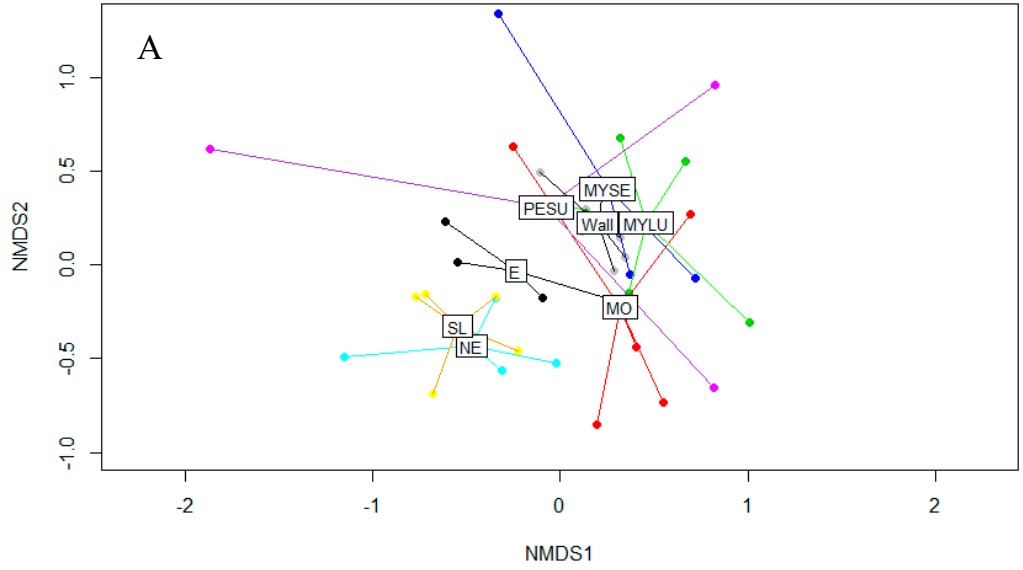

**Figure 3.** *Cont.*

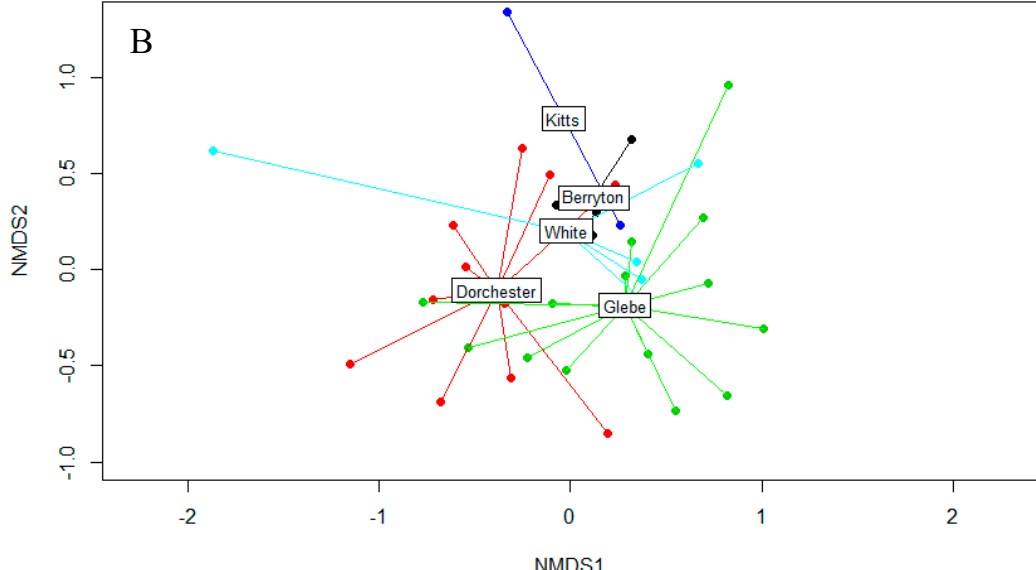

**Figure 3.** Non-metric dimensional scaling of Bray–Curtis dissimilarity index of fungal diversity on different substrates (**A**) and in different underground sites (**B**) in New Brunswick (stress = 0.135). PESU = *Perimyotis subflavus* (magenta in A), MYSE = *Myotis septentrionalis* (blue in A), MYLU = *Myotis lucifugus* (green in A), MO = *Meta ovalis* (red in A), E = *Exechiopsis* spp. (black in A), SL = *Scoliopteryx libatrix* (yellow in A), NE = *Nelima elegans* (cyan in A), wall (grey in A). Colors in (**B**) denote sites: red=Dorchester Mine, green=Glebe Mine, cyan=White Cave, blue=Kitts Cave, and black=Berryton Cave.

**Table 1.** Results of indicator species analysis with adjusted *p*-values by substrate category. Fungi were isolated from multiple species of arthropods and bats in New Brunswick caves, as well as cave walls. Fungal taxa with *p*-values <0.1 are reported. Inval = indicator species statistic.

| Fungal Taxa | Arthropod | Bat | Wall | Inval | *p*-Value |
|---|---|---|---|---|---|
| *Acrodontium* spp. | yes | no | no | 0.594 | 0.087 |
| *Cladosporium* spp. | yes | no | no | 0.798 | 0.054 |
| *Verticillium* sp. (cf. *Gabarnaudia*) | yes | no | no | 0.753 | 0.054 |
| *Mortierella* spp. | no | no | yes | 0.836 | 0.054 |
| *Oidiodendron truncatum* | no | no | yes | 0.877 | 0.054 |
| *Phialophora* spp. | no | no | yes | 0.612 | 0.087 |
| *Pseudogymnoascus pannorum* | no | yes | yes | 0.877 | 0.054 |
| *Trichosporiella* spp. | no | no | yes | 0.800 | 0.054 |
| *Trichosporon* spp. | no | no | yes | 0.637 | 0.054 |
| *Apiotrichum dulcitum* | no | no | yes | 0.802 | 0.054 |

*3.2. Fungal Diversity on Cave Walls*

We cultured a total of 63 fungal taxa in 48 genera from cave walls, with 36.5% of taxa isolated once (Table 2). The most common fungi cultured were similar between years (Table 3). We previously published the detection of *Pd* from cave walls [18]. The Simpson index significantly differed among media types ($F_{3,111}$ = 7.914, $p$ = 7.86 × $10^{-5}$) and site ($F_{4,111}$ = 5.546, $p$ = 0.0004), but not year sampled ($F_{1,111}$ = 0.051, $p$ = 0.821). Dorchester Mine and Kitts Cave had the lowest diversity compared to other sites, and DPYA detected higher fungal diversity than MEA and SD (Figure 4, Table 4).

The composition of fungal assemblages was significantly different among media types (F.model$_{3,110}$ = 5.396, $R^2$ = 0.107, $p$ = 0.001) and sites (F.model$_{4,110}$ = 6.003, $R^2$ = 0.159, $p$ = 0.001), but did not differ over time (F.model$_{1,110}$ = 1.099, $R^2$ = 0.007, $p$ = 0.344). Site had significant dispersion ($F_{4,114}$ = 6.829, $p$ = 0.001), but media type ($F_{3,115}$ = 1.435, $p$ = 0.254) and year did not ($F_{2,116}$ = 0.083, $p$ = 0.922). Sites geographically close together also clustered together: from west to east White Cave

and Berryton Cave (15 km apart), and Glebe Mine and Kitts Cave (16 km apart; Glebe Mine is 46 km from Berryton Cave; Figure 5A). Dorchester Mine is separated from the other sites by two rivers (16 km west from the closest site, White Cave) and clustered separately. However, the significant dispersion of sites resulted in considerable overlap of samples among different sites, and longitude was not significant when site was included in the model first.

Some fungal taxa were significantly associated with specific sites, such as *Phaeotrichum hystricinum* with Glebe Mine (inval stat = 0.567, *p* = 0.03), *Wardomyces giganteus* (indval stat = 0.550, *p* = 0.03) and *Leuconeurospora capsici* (inval stat = 0.728, *p* = 0.03) with Berryton Cave, *Chrysosporium* spp. with Glebe Mine and White Cave (indval stat = 0.482, *p* = 0.03), and *Leuconeurospora polypaeciloides* with Glebe Mine, Kitts Cave, and White Cave (inval stat = 0.705, *p* = 0.03). The fungal diversity cultured from MEA and SD were similar while DPYA and modified DPYA clustered separately (Figure 5B). The modification to the formulation for DPYA did not change the fungal diversity detected. Some fungal taxa were detected significantly more often on some media types, such as *Cephalotrichum stemonitis* (inval stat = 0.680, *p* = 0.04), *Acaulium caviariformis* (inval stat = 0.519, *p* = 0.04), *Pd* (inval stat = 0.675, *p* = 0.04), and *Wardomyces* spp. (inval stat = 0.456, *p* = 0.04) on DPYA and modDPYA, and *Mortierella* spp. on MEA, SD, and modDPYA (inval stat = 0.741, *p* = 0.04). There were no significant indicator species for year.

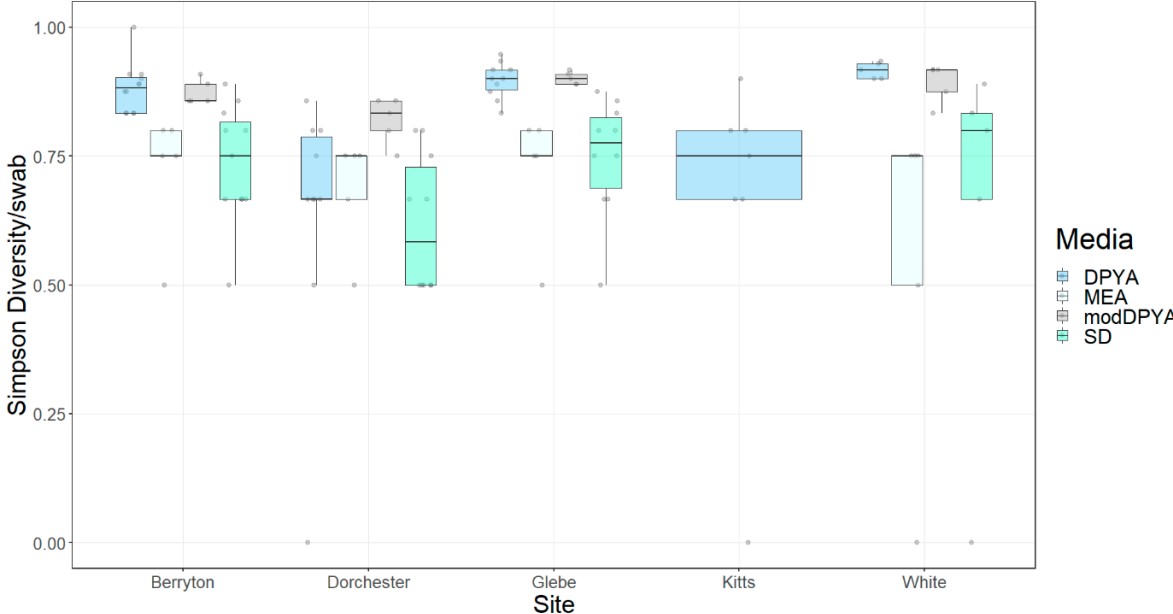

**Figure 4.** Simpson diversity index per swab (represented with dots) on each media type in each site from cave walls in New Brunswick. DPYA = dextrose peptone yeast agar, MEA = malt agar, modDPYA = modified DPYA, and SD = Sabouraud dextrose agar. The boxes represent the mean, the first and third quartiles, and the whiskers extend from the box to the value no further than ± 1.5 multiplied by the inter-quartile range from the boxes.

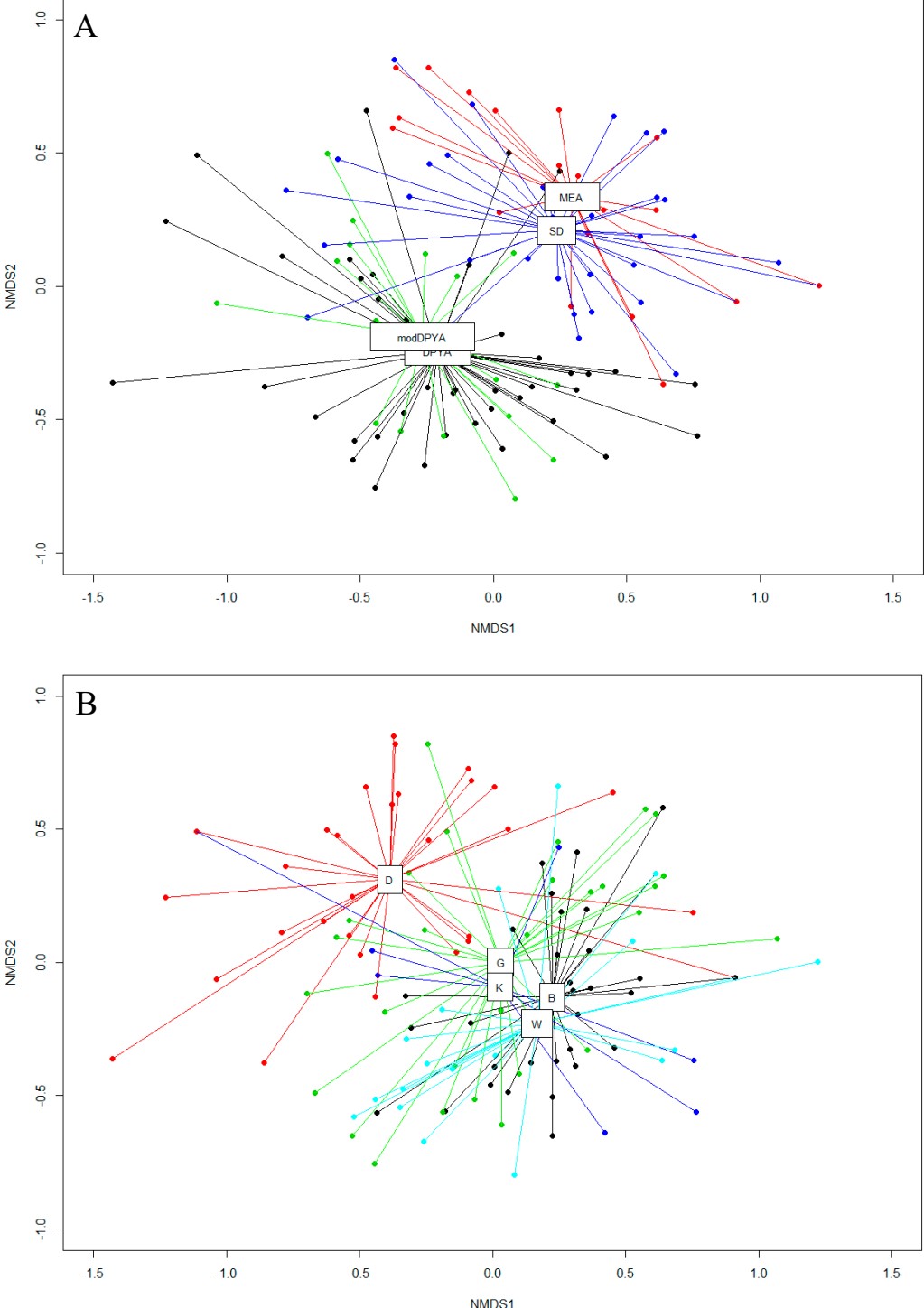

**Figure 5.** Non-metric dimensional scaling of Bray–Curtis dissimilarity index of fungal diversity on cave walls among media types (**A**) and sites (**B**) in New Brunswick (stress = 0.160). (**A**) DPYA = dextrose peptone yeast agar in black, modDPYA = modified DPYA in green, MEA = malt agar in red, and SD = Sabouraud dextrose agar in blue. (**B**) D = Dorchester Mine in red; G = Glebe Mine in green; K = Kitts Cave in blue; B = Berryton Cave in black; W = Whites Cave in cyan.

**Table 2.** Fungal taxa isolated from walls in caves and mines in New Brunswick, Canada in 2012 and 2015. Kitts Cave was sampled in 2013. Column figures indicate the number of swabs culturing positive for each fungal taxon.

| Total Number of Swabs Done | Glebe | Berryton | Dorchester | White | Kitts |
|---|---|---|---|---|---|
| | 30 | 32 | 30 | 20 | 7 |
| **Ascomycota** | | | | | |
| *Acaulium caviariformis* (Malloch and Hubart) Sandoval-Denis, Guarro and Gene | 6 | 5 | 2 | 6 | 0 |
| *Acremonium* sp. | 0 | 1 | 5 | 1 | 1 |
| *Acremonium rutilum* W. Gams | 0 | 0 | 1 | 0 | 0 |
| *Acrodontium* sp. | 1 | 0 | 0 | 0 | 0 |
| *Aphanocladium album* (Preuss) W. Gams | 0 | 0 | 1 | 0 | 0 |
| *Aphanocladium* sp. | 0 | 0 | 1 | 0 | 0 |
| *Arachniotus ruber* (Tiegh.) J. Schröt. | 0 | 6 | 0 | 1 | 0 |
| *Arthroderma silverae* Currah, S.P. Abbott and Sigler | 1 | 4 | 0 | 3 | 0 |
| *Arthrographis* sp. | 0 | 0 | 0 | 1 | 0 |
| *Beauveria* sp. | 2 | 0 | 0 | 0 | 0 |
| *Cadophora* sp. | 5 | 0 | 0 | 0 | 1 |
| *Cephalotrichum* sp. | 0 | 1 | 0 | 0 | 0 |
| *Cephalotrichum stemonitis* (Pers.) Link | 13 | 10 | 3 | 11 | 2 |
| *Chaetomidium* sp. | 0 | 0 | 0 | 1 | 0 |
| *Chalara* sp. | 1 | 0 | 0 | 0 | 0 |
| *Chrysosporium* sp. | 7 | 1 | 1 | 6 | 0 |
| *Cladosporium* sp. | 0 | 3 | 1 | 1 | 0 |
| *Cordyceps* sp. | 0 | 0 | 1 | 1 | 0 |
| *Culicinomyces* sp. | 1 | 0 | 0 | 0 | 0 |
| *Cylindrocarpon* sp. | 0 | 0 | 0 | 0 | 3 |
| *Fusarium* sp. | 3 | 2 | 1 | 1 | 0 |
| *Gymnoascus reesii* Baran. | 1 | 0 | 0 | 0 | 0 |
| *Humicola* cf. UAMH 11595 | 8 | 6 | 0 | 11 | 0 |
| *Hyphozyma* sp. | 1 | 0 | 0 | 0 | 0 |
| *Isaria* sp. | 1 | 0 | 0 | 1 | 0 |
| *Lecythophora* sp. | 1 | 0 | 0 | 0 | 0 |
| *Leuconeurospora polypaeciloides* Malloch, Sigler and Hambleton | 14 | 0 | 0 | 14 | 1 |
| *Leuconeurospora capsici* (J.F.H. Beyma) Malloch, Sigler and Hambleton | 0 | 20 | 0 | 1 | 0 |
| *Mammaria* sp. | 0 | 0 | 0 | 1 | 0 |
| *Microascus* sp. | 1 | 0 | 0 | 0 | 0 |
| *Oidiodendron* sp. | 1 | 0 | 6 | 1 | 0 |
| *Oidiodendron truncatum* G.L. Barron | 7 | 17 | 12 | 11 | 2 |
| *Paecilomyces* sp. | 1 | 2 | 1 | 0 | 0 |
| *Penicillium* spp. | 12 | 28 | 7 | 12 | 4 |
| *Penicillium chrysogenum* | 1 | 0 | 0 | 0 | 0 |
| *Penicillium glaucoalbidum* (Desmazieres) Houbraken and Samson | 1 | 0 | 2 | 0 | 0 |
| *Penicillium griseofulvum* | 0 | 1 | 0 | 0 | 0 |
| *Penicillium thomii* Maire | 1 | 0 | 0 | 0 | 0 |
| *Phaeoacremonium* sp. | 1 | 0 | 0 | 0 | 0 |
| *Phaeotrichum hystricinum* Cain and M.E. Barr | 12 | 0 | 0 | 3 | 0 |
| *Phialophora* sp. | 3 | 0 | 0 | 0 | 1 |
| *Phoma radicina* (McAlpine) Boerema | 1 | 0 | 0 | 0 | 0 |
| *Preussia* sp. | 3 | 0 | 0 | 1 | 0 |
| *Pseudogymnoascus pannorum* sensu lato (Link) Minnis and D.L. Lindner | 12 | 28 | 15 | 11 | 5 |
| *Pseudogymnoascus destructans* (Blehert and Gargas) Minnis and D.L. Lindner | 13 | 8 | 10 | 7 | 0 |
| *Pseudogymnoascus roseus* Raillo | 1 | 0 | 0 | 8 | 0 |
| *Thysanophora* sp. | 2 | 0 | 0 | 0 | 0 |
| *Tolypocladium* sp. | 3 | 0 | 2 | 1 | 1 |
| *Tricellula* cf. *aquatica* | 0 | 0 | 0 | 0 | 2 |
| *Trichoderma* sp. | 1 | 3 | 0 | 0 | 1 |
| *Trichophyton* sp. | 0 | 0 | 1 | 0 | 0 |
| *Trichosporiella* sp. | 2 | 1 | 1 | 1 | 1 |
| *Verticillium* sp. | 2 | 1 | 1 | 0 | 0 |
| *Verticillium* sp. cf. *Gabarnaudia* | 0 | 1 | 0 | 0 | 1 |

**Table 2.** *Cont.*

| Total Number of Swabs Done | Glebe | Berryton | Dorchester | White | Kitts |
|---|---|---|---|---|---|
| | 30 | 32 | 30 | 20 | 7 |
| *Wardomyces* spp. | 6 | 2 | 0 | 2 | 0 |
| *Wardomyces giganteus* (Malloch) Sandoval-Denis, Guarro and Gene | 0 | 10 | 0 | 0 | 0 |
| *Wardomyces inflatus* (Marchal) Hennebert | 2 | 1 | 0 | 0 | 0 |
| *Zalerion* sp. | 0 | 0 | 1 | 0 | 0 |
| **Basidiomycota** | | | | | |
| *Apiotrichum dulcitum* (Berkhout) Yurkov and Boekhout | 12 | 6 | 0 | 1 | 3 |
| *Asterotremella* sp. | 1 | 0 | 0 | 2 | 0 |
| *Trichosporon* sp. | 6 | 2 | 0 | 2 | 0 |
| **Mucoromycota** | | | | | |
| *Thamnidium elegans* Link | 1 | 0 | 0 | 2 | 0 |
| *Mortierella* spp. | 20 | 9 | 16 | 5 | 1 |
| *Mucor* spp. | 15 | 21 | 3 | 12 | 2 |
| **Zoopagomycota** | | | | | |
| *Kickxella alabastrina* Coem. | 1 | 0 | 0 | 0 | 0 |
| Sterile | 9 | 2 | 15 | 7 | 1 |

**Table 3.** The most common fungal taxa cultured from cave walls in New Brunswick in each year. Numbers indicate the percentage of samples which were positive for the indicated taxa. *n* = 32 in 2012 and *n* = 80 in 2015.

| Fungal Taxa | 2012 | 2015 |
|---|---|---|
| *Pseudogymnoascus pannorum s.l.* | 59.4 | 58.8 |
| *Penicillium* spp. | 50.0 | 53.8 |
| *Mortierella* spp. | 43.8 | 45.0 |
| *Pseudogymnoascus destructans* | 40.6 | 31.3 |
| *Mucor* spp. | 37.5 | 48.8 |
| *Oidiodendron truncatum* | 31.3 | 46.3 |
| *Cephalotrichum stemonitis* | 28.1 | 35.0 |
| *Leuconeurospora polypaeciloides* | 18.8 | 27.5 |

**Table 4.** The mean number of fungal taxa cultured per swab ± standard deviation for each year in each site. Kitts Cave was sampled once, in 2013, with a mean of 4.7 ± 2.9 fungal taxa cultured per swab and 17 fungal genera detected (*n* = 7 swabs). DPYA = dextrose peptone yeast agar. MEA = malt agar. SD = Sabouraud dextrose agar. ND = no data. # = number of.

| Site/Medium | # Fungal Genera | # Fungal Taxa/Swab 2012 | # Fungal Taxa/Swab 2015 | Overall Mean |
|---|---|---|---|---|
| Glebe Mine | 38 | 6.3 ± 3.2, *n* = 10 | 7.8 ± 4.2, *n* = 20 | 7.3 ± 3.9, *n* = 30 |
| Berryton Cave | 22 | 4.9 ± 2.8, *n* = 12 | 6.5 ± 2.9, *n* = 20 | 6.1 ± 2.9, *n* = 32 |
| Dorchester Mine | 20 | 3.6 ± 1.8, *n* = 10 | 3.7 ± 1.7, *n* = 20 | 3.7 ± 1.7, *n* = 30 |
| White Cave | 28 | ND | 7.5 ± 4.5, *n* = 20 | 7.5 ± 4.5, *n* = 20 |
| SD | 26 | 3.6 ± 1.2, *n* = 16 | 4.7 ± 2.5, *n* = 20 | 4.2 ± 2.1, *n* = 36 |
| MEA | 16 | ND | 3.6 ± 1.2, *n* = 20 | 3.6 ± 1.2, *n* = 20 |
| DPYA | 40 | 7.3 ± 3.3, *n* = 16 | 8.5 ± 4.9, *n* = 20 | 7.7 ± 4.2, *n* = 36 |
| modDPYA | 32 | ND | 8.6 ± 2.5, *n* = 20 | 8.6 ± 2.5, *n* = 20 |
| Total/Overall Mean | 48 | 5.1 ± 2.8, *n* = 32 | 6.4 ± 3.8, *n* = 80 | 6.0 ± 3.6, *n* = 112 |

## 4. Discussion

Substrate was the major factor explaining variance in fungal diversity among samples from bats, arthropods, and cave walls, followed by site. Glebe Mine and Dorchester Mine were distinct from the other sites likely because these were the only sites where arthropods were sampled. Cave walls had the highest fungal diversity while overall bats had the lowest. Fungal diversity on cave walls in China was also higher compared to other cave substrates, such as sediments and air [6].

However, while [37] found that bacterial diversity was higher on cave walls compared to bats, fungal diversity was similar. This difference in results may be due to differing methodologies, since the culture-independent methods used by [37] detects both viable and non-viable fungi. Fungi present on cave walls are likely introduced by air currents, percolating water, and cave fauna and fungi may also grow on the rock surface [3–5]. This could account for the relatively high fungal diversity on cave walls compared to bats and arthropods, which may decrease fungal diversity on their external surfaces through grooming or be unsuitable substrates for fungal growth. Cave fauna that physically contact cave walls may acquire a subset of the fungal diversity on walls, as well as being contributors. Unlike hibernating bats, arthropods such as *Meta ovalis*, *Nelima elegans*, and *Exechiopsis* spp., but not *Scoliopteryx libatrix*, are active all winter, likely interacting with a greater variety of environmental substrates, and therefore acquiring a higher fungal diversity on their external surfaces. Arthropods had a higher diversity of fungal pathogens specific to insects and low occurrence of *Pseudogymnoascus pannorum* on their surface compared to bats and cave walls.

We did not detect changes in the culturable fungal assemblage present on cave walls with the introduction of *Pd* to the region and the subsequent disappearance of the hibernating bat population over a 3-year period. *Pd* co-exists with a similar diversity of fungi on cave walls in New York [38], and *Pd* has also co-existed with a wide diversity of fungi in Eurasian caves for hundreds to thousands of years [2,39]. However, except for Glebe Mine (in which we first observed *Pd* growth on bats concurrent with wall swabbing in 2012), we first sampled our sites (in 2012) 3 months to a maximum of 1 year after we first detected *Pd* at each site [18], so the possibility that changes in the fungal assemblage had already occurred cannot be ruled out. Additionally, microbial assemblages that were not cultured by our methods, such as yeasts and bacteria, may have changed over time. Nevertheless, by winter 2015 the hibernating bat population observed in the sampled sites totaled 6 individuals, reduced from 435 winter 2012 (maximum of 6450 bats counted in these sites winter 2011; [20,22], unpublished data), yet fungal diversity on cave walls increased during that time, although the trend was not significant. This suggests that fungi documented in caves prior to the outbreak of *Pd* do not require regular transmission of spores by bats to maintain fungal diversity at these sites, although bats may be important for the initial introduction of a fungus, such as *Pd*. It may require a longer time period for the effects of the absence of bats and the introduction of *Pd* on the fungal assemblage in caves to be measurable, if any change occurs.

Media type was the best explanatory variable for the difference in cave wall fungal diversity among samples, while site was the best explanatory variable for fungal composition. Isolates from DPYA represented a higher fungal diversity than those from either SD or MEA, and the fungal composition also differed. Slow growing fungi, such as *Acaulium caviariformis*, were more often cultured on DPYA compared to MEA and SD, which tended to be overgrown with faster-growing fungi such as *Mortierella* spp. that thrive in the high-sugar conditions of those media types. Aside from low sugar, DPYA also contains ox-gall (selective against some bacteria and a wetting agent) and sodium propionate (inhibits fungi). Using additional media types would undoubtedly detect additional fungal diversity. Autoclaving ingredients separately in making DPYA did not change the results. Therefore, the bioproducts generated in autoclaving agar and phosphate buffer together do not appear to inhibit the detection of fungal species in caves, though it does so with microbes in other habitats [31]. Culture-dependent surveys should use multiple media types that differ in their sugar concentrations, microbial inhibitors (such as sodium propionate and ox-gall in DPYA), and other nutrients to detect maximal fungal diversity.

Dorchester Mine had the lowest fungal diversity on cave walls of all sampled sites. Dorchester Mine is an abandoned copper mine and has visible deposits of copper on the walls and elevated copper levels in the water compared to the other sites [40]. Copper can inhibit fungal growth and is an active component in many fungicides [41]. Additionally, Dorchester Mine is the only site where the cave floor is entirely covered by a slow-moving stream. In [21], it was found that fungal diversity on hibernating bats was lowest at sites that had significant water bodies, which may limit the availability of organic

matter in caves or inhibit spore germination. Dorchester Mine has no known mammalian residents, aside from hibernating bats and deer mice (*Peromyscus maniculatus*). This is unlike Glebe Mine, Kitts Cave, and White Cave, where porcupines (*Erethizon dorsatum*) or beavers (*Castor canadensis*) introduce significant quantities of organic matter (dung and vegetation) which may increase fungal diversity. In [42], it was found that on Anticosti Island, Quebec, a lack of mammals associated with caves was correlated with the absence of some fungi. Organic matter on the floor of caves may also produce gases that serve as nutrition sources for microbes on cave walls [9]. Species such as *Leuconeurospora* spp., *Cephalotrichum stemonitis*, *Trichosporon* spp., and *Phaeotrichum hystricinum* appear to be associated with dung in caves [43,44] and were not detected in Dorchester Mine. In [1], it was found that arthropod richness and fungal richness in cave substrates was positively correlated, however in our study Dorchester Mine, along with Glebe Mine, had the greatest number of macro-invertebrates compared to the other sites sampled [20], yet had the lowest fungal diversity on cave walls. Fungal colonies were visible on the cave walls of Kitts Cave more so than any other site, which may be related to the thick clay mud covering some of the walls in that site deposited by seasonal flooding.

Avenues for future research include the ecological interactions and modes of nutrition for fungi, including *Pd*, on cave walls. Caves with large dung deposits, particularly those with large bat colonies [14], may have unique microbial assemblages suited to extracting nutrients from gases, such as ammonia, in the cave atmosphere.

**Author Contributions:** Conceptualization, K.J.V., D.M., D.F.M.; methodology, K.J.V., D.M., D.F.M.; software, K.J.V.; formal analysis, K.J.V.; investigation, K.J.V., D.M., D.F.M.; resources, K.J.V., D.M., D.F.M.; data curation, K.J.V.; writing—original draft preparation, K.J.V.; writing—review and editing, K.J.V., D.M., D.F.M.; visualization, K.J.V.; supervision, D.M., D.F.M.; project administration, K.J.V., D.F.M.; funding acquisition, D.F.M.

**Funding:** This research was funded by the Crabtree Foundation, New Brunswick Environmental Trust Fund, New Brunswick Wildlife Trust Fund, New Brunswick Department of Natural Resources, and Parks Canada.

**Acknowledgments:** Access to hibernacula on private lands was generously provided by D. Roberts, J. Chown, and T. Gilchrist. Scientific permits for entering sites were provided by the New Brunswick Department of Natural Resources Species-at-Risk Program and the New Brunswick Protected Natural Areas Program.

**Conflicts of Interest:** The authors declare no conflict of interest. The funders had no role in the design of the study; in the collection, analyses, or interpretation of data; in the writing of the manuscript, or in the decision to publish the results.

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
