# Peer review of "No Change Detected in Culturable Fungal Assemblages on Cave Walls in Eastern Canada with the Introduction of Pseudogymnoascus destructans"

_diversity, doi:10.3390/d11120222_

Round 1

Reviewer 1 Report

No change detected in culturable fungal assemblages on cave walls in Eastern Canada with the introduction of Pseudogymnoascus destructans

Karen J Vanderwolf, David Malloch and Donald F. McAlpine

The diversity of cave wall fungi was accessed using culture dependent techniques. Previous fungal assemblages (mite and bats) were also analyzed alongside the new information.

Strengths:

1) The article is unique in that they have assemblages from cave walls, mites, and bats from the same time period, using the same techniques.

2) The use of multiple media for isolation.

Introduction:

The introduction could be restructured slightly to make it flow better. The first paragraph does not mention bats or WNS so later paragraphs appear a little disconnected. The second paragraph discusses excellent information why walls might have a high fungal diversity and why they might not rely on bats but perhaps this information would be best moved to the discussion as a justification or hypothesis as to why wall diversity has not changed? In addition, some of this information would be good to suggest for future study in a conclusion paragraph.

Specific comments:

Line 33: Fungal spores can be transported into caves by water, wind, and fauna.

Seem out of place at the end of the paragraph. Perhaps insert this sentence after the sentence in line 29 and alter the following sentence.

example: [1,2] Fungal spores can be transported into caves by water, wind , and fauna. In fact, the majority of fungi documented from caves appear to originate.

Line 50: or by oxidizing metals within rocks.

Are all metals bound to rocks? Fungi could also oxidize metal ions in solution so perhaps shorten to

"or by oxidizing metals."

Line 62: Word choice, " the fungal assemblage on the rock." This fungal assemblage is associated with cave walls so the clay wall hibernacula (Figure 1) would not apply in terms of rock. I would suggest consistent word choice for the cave wall fungal assemblage and to caution introducing an additional term that could confuse the reader.

Line 65: Please remove (and hence living)... this is not needed. Also, the hard-to-culture and the not-yet-cultured fungal groups are also living.

Materials and Methods:

 Line 73: The mean winter..........[25]. This sentence is not needed since this information is represented within the table cited in the previous sentence. My initial impression was that this temperature reference was to justify the incubation temperature but further along the incubation temperature was 7°C  (not 5.1 (4.0-6.1).

Line 74: We followed the protocol of [27] for minimizing the spread of WNS during all visits to caves.

Important sentence but please move if to the very last sentence after explaining your sampling method. Appears out of place in this location.

Lines 82: (unpublished data). Not sure if you need this.                                         

Line 91: ....deep in hibernacula.......  deep is a hard word to replicate. Can you state this in another way that other researchers could go to a similar area based on the description.

Line 107: [31] found that autoclaving...... The reference is related to bacteria not fungi so you need to make the connection to the reader why it's important to your study.

Line 116-119: If you utilized all these resource just state it. The word choice is passive and when you state 116 (We also had access to....) and 119 (usually through a combination....) it makes the reader wonder if you used the reference collections and how many were un-usual.

example: Identifications were carried out by comparing the micro- and maromorphological characteristics of the microfungi to the taxonomic literature and compendia [32] and using a reference collection of identified cultures obtained from Myotis spp. in 2010. Isolates with unresolved identification were sent to taxonomic specialists.

Lines 146-147: NMDS plots of Jaccard distance metrics were constructed to visualize hoe different sites and media types cluster. Figure 5 shows media and sites but the legend stated bray-curtis. Although it is stated that NMDS plots using Bray-Curtis were made from averaged data (lines 136-137). Is 5A and 5B, really Bray-Curtis or are they Jaccard? If Figure 5 is Bray-Curtis, mentioning NMDS plots of Jaccard distance is not shown, confusing to the reader, and is not needed.

Results:

Line 159: Sites tended not to cluster...tended (either they did not not).

Line 161: likely because only two sites where arthropods were sampled. This is an interpretation of the result which is generally confined to the discussion, but it depends on the journal.

Line 197: The fungal diversity detected by MEA and SD were similar.... The media does not detect fungi. Please re-word. ......

Line 199: DPYA did not appear to change the fungal diversity detected. Same comment as above and did not appear is passive (did it change or not?).

Discussion:

Line 248-252: Media does not detect. Also, beyond sugar amount. DPYA has oxgall and sodium propionate. Ox-gall is selective against some bacteria and a wetting agent, while sodium propionate inhibits fungi so this is probably more important/equally important as sugar content to why different communities were found. Wetting agents increase water availability (good for some fungi while detrimental to others).

Lines 253-254: Autoclaving ingredients separately in making DPYA did increase the diversity detected, but not significantly. The media does not detect as previously mentioned. Also an increase that is not statistically significant is hard to state when isolation is based on picking unique colonies vs all the colonies. This might statement is likely a stretch.

Lines 259: The variation among sites in fungal diversity appeared to included a geographic component, as sites located close together were also more similar.....  You can test if geographical distance accounts for a large part of the variation using NMDS bray-Curtis ordination and envfit with location as meta-data in R. I would also do ANOSIM to see if the results are similar.

#all useful packages, install.packages(...package name...)

library(lmerTest)

library(bbmle)

library(reshape)

library(vegan)

library(ggplot2)

library(plyr)

library(ca)

library(MASS)

library (vegan3d)

library(geometry)

library(rgl)

library(mvabund)

library(vegan)

library(GUniFrac)

library(ade4)

library(corrplot)

library(picante)

######Using a different data set and Metadata

OTUdata<-read.csv(file.choose(), row.names=1,check.names=FALSE)

Metadata<-read.csv(file.choose(), check.names=FALSE)

############ NMDS showing habitat

OTU2 <-vegdist(OTUdata, method="bray", binary=FALSE, diag=FALSE, upper=FALSE,  na.rm = FALSE)

OTU2

test <- metaMDS(OTU2,k=9,itr=20)

NMDS=test

attach(Metadata)

plot(NMDS, disp="sites", type="n")

palette(c("coral","brown2","lightblue","blue3","blue4","orange"))

ordiellipse(NMDS, Location, col=1:6, kind = "ehull", lwd=3)

ordiellipse(NMDS, Location, col=1:6, draw="polygon")

points(NMDS, disp="site", pch=21, col="Black", bg="Black", cex=1)

orditorp(NMDS, disp="site", pch=3, col="Black", bg="Black", cex=1)

ord.fit <- envfit(OTU2 ~ Location, data=Metadata, perm=1000)

ord.fit

Figures and tables:

Figure 5.

Is this Bray-curtis or Jaccard? Lines 146-147: NMDS plots of Jaccard distance metrics were constructed to visualize hoe different sites and media types cluster. Figure 5 shows media and sites but the legend stated bray-curtis. Although it is stated that NMDS plots using Bray-Curtis were made from averaged data (lines 136-137). Is 5A and 5B, really Bray-Curtis or are they Jaccard? If Figure 5 is Bray-Curtis, mentioning NMDS plots of Jaccard distance is not shown, confusing to the reader, and is not needed.

Table 2.

Please reconsider the use of rock walls. Are they all bare rock walls or rock covered in clay? Why not just cave walls?

Please review the taxonomic determination. Kickxella alabastrina under your Ascomycota column. I did not check them all but should be reviewed .

Kickxella alabastrina Coem., Bull. Soc. R. Bot. Belg. 1: 156 (1862)

Kickxellaceae, Kickxellales, Incertae sedis, Kickxellomycetes, Kickxellomycotina, Zygomycota, Fungi

Author Response

Introduction:

The introduction could be restructured slightly to make it flow better. The first paragraph does not mention bats or WNS so later paragraphs appear a little disconnected. The second paragraph discusses excellent information why walls might have a high fungal diversity and why they might not rely on bats but perhaps this information would be best moved to the discussion as a justification or hypothesis as to why wall diversity has not changed? In addition, some of this information would be good to suggest for future study in a conclusion paragraph.

-some changes made to Intro to improve flow and introduce reference to bats in the opening paragraph. The relevant section of the discussion has also been clarified. Generally, we have not made major changes to the Introduction as we feel this information is important to introduce at the beginning, and Reviewer 2 seemed satisfied with it. We have added text to the concluding paragraph of the discussion regarding future study directions. 

Specific comments:

Line 33: Fungal spores can be transported into caves by water, wind, and fauna.

Seem out of place at the end of the paragraph. Perhaps insert this sentence after the sentence in line 29 and alter the following sentence.

example: [1,2] Fungal spores can be transported into caves by water, wind , and fauna. In fact, the majority of fungi documented from caves appear to originate.

-done as suggested

Line 50: or by oxidizing metals within rocks.

Are all metals bound to rocks? Fungi could also oxidize metal ions in solution so perhaps shorten to "or by oxidizing metals."

-changed to “or by oxidizing metals in the environment.”

Line 62: Word choice, " the fungal assemblage on the rock." This fungal assemblage is associated with cave walls so the clay wall hibernacula (Figure 1) would not apply in terms of rock. I would suggest consistent word choice for the cave wall fungal assemblage and to caution introducing an additional term that could confuse the reader.

-changed to “the fungal assemblage on these areas”

Line 65: Please remove (and hence living)... this is not needed. Also, the hard-to-culture and the not-yet-cultured fungal groups are also living.

-done as suggested

Materials and Methods:

 Line 73: The mean winter..........[25]. This sentence is not needed since this information is represented within the table cited in the previous sentence. My initial impression was that this temperature reference was to justify the incubation temperature but further along the incubation temperature was 7°C  (not 5.1 (4.0-6.1).

-sentence removed as suggested

Line 74: We followed the protocol of [27] for minimizing the spread of WNS during all visits to caves.

Important sentence but please move if to the very last sentence after explaining your sampling method. Appears out of place in this location.

-done as suggested

Lines 82: (unpublished data). Not sure if you need this. 

-it has been removed                                       

Line 91: ....deep in hibernacula.......  deep is a hard word to replicate. Can you state this in another way that other researchers could go to a similar area based on the description.

-replaced with “in the dark zone of hibernacula in areas where bats were known to routinely roost pre-WNS”

Line 107: [31] found that autoclaving...... The reference is related to bacteria not fungi so you need to make the connection to the reader why it's important to your study.

-changed to “found that autoclaving ingredients separately increased the number of bacterial species that could be cultured, and this could potentially apply to culturable fungal diversity as well.”

Line 116-119: If you utilized all these resource just state it. The word choice is passive and when you state 116 (We also had access to....) and 119 (usually through a combination....) it makes the reader wonder if you used the reference collections and how many were un-usual.

example: Identifications were carried out by comparing the micro- and maromorphological characteristics of the microfungi to the taxonomic literature and compendia [32] and using a reference collection of identified cultures obtained from Myotis spp. in 2010. Isolates with unresolved identification were sent to taxonomic specialists.

- the methods section has been changed from passive voice to active voice. The sentence referenced above has been changed to: “We identification isolates by comparing micro- and macromorphological characteristics of the microfungi to taxonomic literature and compendia [32,33], and by using a reference collection of cultures from Myotis spp. collected in 2010 that were previously identified using a mix of sequencing and morphological features [21].”

Lines 146-147: NMDS plots of Jaccard distance metrics were constructed to visualize hoe different sites and media types cluster. Figure 5 shows media and sites but the legend stated bray-curtis. Although it is stated that NMDS plots using Bray-Curtis were made from averaged data (lines 136-137). Is 5A and 5B, really Bray-Curtis or are they Jaccard? If Figure 5 is Bray-Curtis, mentioning NMDS plots of Jaccard distance is not shown, confusing to the reader, and is not needed.

-lines 146-147 were incorrect and have been changed. The figure 5 caption was correct. Thank you for detecting this mistake!

Results:

Line 159: Sites tended not to cluster...tended (either they did not not).

-revised to "did not cluster"

Line 161: likely because only two sites where arthropods were sampled. This is an interpretation of the result which is generally confined to the discussion, but it depends on the journal.

-this has been moved to the discussion

Line 197: The fungal diversity detected by MEA and SD were similar.... The media does not detect fungi. Please re-word. ......

-text has been reworded

Line 199: DPYA did not appear to change the fungal diversity detected. Same comment as above and did not appear is passive (did it change or not?).

-text has been reworded

Discussion:

Line 248-252: Media does not detect. Also, beyond sugar amount. DPYA has oxgall and sodium propionate. Ox-gall is selective against some bacteria and a wetting agent, while sodium propionate inhibits fungi so this is probably more important/equally important as sugar content to why different communities were found. Wetting agents increase water availability (good for some fungi while detrimental to others).

-this information has been added to the discussion: “Aside from low sugar, DPYA also contains ox-gall (selective against some bacteria and a wetting agent) and sodium propionate (inhibits fungi).”

Lines 253-254: Autoclaving ingredients separately in making DPYA did increase the diversity detected, but not significantly. The media does not detect as previously mentioned. Also an increase that is not statistically significant is hard to state when isolation is based on picking unique colonies vs all the colonies. This might statement is likely a stretch.

-this sentence has been changed to “Autoclaving ingredients separately in making DPYA did not change the results.”

Lines 259: The variation among sites in fungal diversity appeared to included a geographic component, as sites located close together were also more similar.....  You can test if geographical distance accounts for a large part of the variation using NMDS bray-Curtis ordination and envfit with location as meta-data in R. I would also do ANOSIM to see if the results are similar.

- The idea of a geographic component has been moved from the discussion and into the results: “Sites geographically close together also clustered together: from west to east White Cave and Berryton Cave (15 km apart), and Glebe Mine and Kitts Cave (16 km apart; Glebe Mine is 46 km from Berryton Cave; Figure 5A). Dorchester Mine is separated from the other sites by two rivers (16 km west from the closest site, White Cave) and clustered separately. However, the significant dispersion of sites resulted in considerable overlap of samples among different sites, and longitude was not significant when site was included in the model first.” The effect of geographic site was tested by adding a variable to the PERMANOVA. The function envfit and an ANOSIM gave the same results as the PERMANOVA (only the latter results are included in the manuscript).

Figures and tables:

Figure 5.

Is this Bray-curtis or Jaccard? Lines 146-147: NMDS plots of Jaccard distance metrics were constructed to visualize hoe different sites and media types cluster. Figure 5 shows media and sites but the legend stated bray-curtis. Although it is stated that NMDS plots using Bray-Curtis were made from averaged data (lines 136-137). Is 5A and 5B, really Bray-Curtis or are they Jaccard? If Figure 5 is Bray-Curtis, mentioning NMDS plots of Jaccard distance is not shown, confusing to the reader, and is not needed.

-lines 146-147 were incorrect and have been changed. The figure 5 caption was correct. Thank you for detecting this mistake!

Table 2.

Please reconsider the use of rock walls. Are they all bare rock walls or rock covered in clay? Why not just cave walls?

-Most are bare rock, one is covered in clay. The title of table 2 has been modified as suggested.

Please review the taxonomic determination. Kickxella alabastrina under your Ascomycota column. I did not check them all but should be reviewed .

Kickxella alabastrina Coem., Bull. Soc. R. Bot. Belg. 1: 156 (1862)

Kickxellaceae, Kickxellales, Incertae sedis, Kickxellomycetes, Kickxellomycotina, Zygomycota, Fungi

-this has been corrected

Reviewer 2 Report

The authors studied an interesting topic, the impact of the decline of bats on the overall diversity of fungi in the underground environment because bats are one of the sources of this diversity. It is an important issue for the protection of the natural environment and the possible protective measures. The main goal is to compare the situation before and after 3 years after the decline of bats.

The manuscript is written concisely. The results are evaluated by statistical methods.

Discussion: The authors are aware that the other isolation media used could increase the fungal diversity. As well as a longer time period could show more in terms of the impact of bat decline on diversity.

I have no serious comments.

The main weaknesses of the work:

The identification of fungi is based solely on phenotype features. However, I understand that the cultivation of fungi from the underground is both time-consuming and they are groups of very few explored fungi.  

Other comments

Table 2:

Penicillium sp., Wardomyces sp., Mortierella sp., and Mucor sp. according to the given positive swabs probably represent several species. Thus they should be mentioned as spp. Penicillium spp. should be then situated behind Penicillium thomii

Kicxella alabastrina and Thamnidium elegans are incorrectly included in the Ascomycota. According to Tedersoo et al. (2019, Fungal Diversity) they belong to Kickxellomycota and Mucoromycota, respectively.

Table 3:

Penicillium sp., Mortierella sp., and Mucor sp. should be probably mentioned with spp.

Regarding the formal level of the article, I found only several minor mistakes.

Incorrect version / correct version

line 118: Meta ovalis / Meta ovalis (italics)

208, 216, and 228: sabouraud / Sabouraud

Table 3: sl. / s.l.

316: Novacova / Nováková

321: Cave : Mount Sedom , The / Cave: Mount Sedom, The (remove the gaps)

326: micobiome / mycobiome

334: “Geomycology : fungi in mineral substrata” is duplicated

342: Geomyces destructans / Geomyces destructans (italics)

349: destructans , the causal / destructans, the causal (remove the gap)

351: 78. / 78: 158-162.

354: 2015. / 2015, 282: 20142335.

359: Myotis / Myotis (italics)

360: Pseudogymnoascus destructans / Pseudogymnoascus destructans (italics)

381: A Hidden Pitfall in the Preparation of Agar Media Undermines. / A hidden pitfall in the preparation of agar media undermines microorganism cultivability. Appl. Environ. Microbiol.

384: Baveria / Bavaria

Author Response

Table 2:

Penicillium sp., Wardomyces sp., Mortierella sp., and Mucor sp. according to the given positive swabs probably represent several species. Thus they should be mentioned as spp. Penicillium spp. should be then situated behind Penicillium thomii

-this has been corrected

Kicxella alabastrina and Thamnidium elegans are incorrectly included in the Ascomycota. According to Tedersoo et al. (2019, Fungal Diversity) they belong to Kickxellomycota and Mucoromycota, respectively.

-this has been corrected

Table 3:

Penicillium sp., Mortierella sp., and Mucor sp. should be probably mentioned with spp.

 -this has been corrected

Regarding the formal level of the article, I found only several minor mistakes.

-all minor mistakes corrected

Incorrect version / correct version

line 118: Meta ovalis / Meta ovalis (italics)

208, 216, and 228: sabouraud / Sabouraud

Table 3: sl. / s.l.

316: Novacova / Nováková

321: Cave : Mount Sedom , The / Cave: Mount Sedom, The (remove the gaps)

326: micobiome / mycobiome

334: “Geomycology : fungi in mineral substrata” is duplicated

342: Geomyces destructans / Geomyces destructans (italics)

349: destructans , the causal / destructans, the causal (remove the gap)

351: 78. / 78: 158-162.

354: 2015. / 2015, 282: 20142335.

359: Myotis / Myotis (italics)

360: Pseudogymnoascus destructans / Pseudogymnoascus destructans (italics)

381: A Hidden Pitfall in the Preparation of Agar Media Undermines. / A hidden pitfall in the preparation of agar media undermines microorganism cultivability. Appl. Environ. Microbiol.

384: Baveria / Bavaria